# Perineural Invasion and Associated Pain Transmission in Pancreatic Cancer

**DOI:** 10.3390/cancers13184594

**Published:** 2021-09-13

**Authors:** Jialun Wang, Yu Chen, Xihan Li, Xiaoping Zou

**Affiliations:** Department of Gastroenterology, Nanjing Drum Tower Hospital, The Affiliated Hospital of Nanjing University Medical School, Nanjing 210008, China; 18362926723@163.com (J.W.); chenyu01_29@163.com (Y.C.); xihanli@nju.edu.cn (X.L.)

**Keywords:** pancreatic cancer, perineural invasion, nerve remodeling, pain

## Abstract

**Simple Summary:**

Perineural invasion is a complicated process involving a series of cells and extracellular matrix components in the tumor microenvironment, particularly the crosstalk between cancer cells and neurons. Perineural invasion occurs in many malignant tumors, including gastric carcinoma, biliary tract tumor and pancreatic cancer. It is identified in approximately 80–100% of pancreatic cancer patients and is correlated with poor survival and decreased quality of life. Extensive studies have revealed the subtle molecule regulatory mechanisms during perineural invasion, as well as the potential causal link with pancreatic cancer-associated pain. Here, we introduce the underlying mechanism of perineural invasion and its possible relationship with the intractable pain in pancreatic cancer patients.

**Abstract:**

Pancreatic ductal adenocarcinoma (PDAC) is one of the cancers with the highest incidence of perineural invasion (PNI), which often indicates a poor prognosis. Aggressive tumor cells invade nerves, causing neurogenic inflammation; the tumor microenvironment also induces nerves to undergo a series of structural and functional reprogramming. In turn, neurons and the surrounding glial cells promote the development of pancreatic cancer through autocrine and/or paracrine signaling. In addition, hyperalgesia in PDAC patients implies alterations of pain transmission in the peripheral and central nervous systems. Currently, the studies on this topic are relatively limited. This review will elaborate on the mechanisms of tumor–neural interactions and its possible relationship with pain from several aspects that have been focused on in recent years.

## 1. Introduction

PDAC is one of the most lethal malignant tumors with a five-year survival rate of less than 10% [1,2]. PNI, a phenomenon defined as cancer cells that surround more than 1/3 of the nerve periphery or invade any layer of the neurolemma [3], occurs in approximately 80–100% of the affected individuals and has become one of the most outstanding clinical features of pancreatic cancer [4].

As more and more clinical studies have shown that PNI is related to poor prognosis [4], the role of nerves in pancreatic cancer has attracted widespread attention over the last few years. As the basis of PNI, the innervation in PDAC has been confirmed, including sensory and autonomic nerves.

The tumor microenvironment and cancer cells of PDAC influence the nervous system in several aspects. As early as the pancreatic intraepithelial neoplasia (PanIN) stage, the alterations of intrapancreatic nerves have been observed, which is the so-called “neural remodeling” [5]. With further deterioration, cancer cells directly contact the nerve cells, promoting the development of neurogenic inflammation in PDAC. The nerve invasion eventually becomes one of the primary sources of PDAC-related pain.

At the level of the central nervous system (CNS), primary tumor cells can secrete inflammatory mediators, increasing immune infiltration into the CNS. This may partially explain the mental symptoms happening in PDAC patients [6].

Nerves are more than “victims” of cancer invasion. Recent studies revealed their role in facilitating neural dissemination. The specific mechanisms are as follows: (i) secreting some chemotactic factors to induce the neutrophilic movement of tumor cells [4,7,8,9,10]; (ii) migrating toward the tumor actively [11,12]; and (iii) increasing the number and length of synapses, thus shortening the physical distance between each other [13].

Although there have been plenty of studies on the “influence of nerves on PNI,” we still know very little about the overall impact of PNI. An in-depth understanding of the mechanisms concerning the phenomenon will help us to better target the tumor–nerve interplay that affects malignant behavior patterns, so as to develop a potentially effective treatment strategy. This article will introduce the intricate regulatory mechanism between PDAC and nerves and its possible connection with pain in detail.

## 2. Intrapancreatic Innervation and Neural Remodeling in PDAC

The pancreas has multiple innervations, including sympathetic, parasympathetic, and sensory nerves. The intrapancreatic sympathetic nerves originate from the celiac ganglia and are evenly distributed in the pancreatic parenchyma [14]. The parasympathetic nerves primarily come from the vagus nerve, of which up to 80% are sensory [15]. Researchers have also confirmed the presence of both myelinated and unmyelinated sensory nerves within the pancreas, using antibodies against neurofilament H (RT97) and calcitonin gene-related peptide (CGRP), which could label the two kinds of nerve fibers, respectively [14].

Nerves are an essential part of the pancreas and are implicated in PDAC patients. Studies have shown that inflammation and neuronal damage of the peripheral and central nervous system have occurred as early as the PanIN stage [5]. At this stage, a significant increase in the sensory innervation of pancreas can be found and are manifested as a decrease in exploratory behavior [16]. In terms of autonomic nerves, while the number of parasympathetic nerves did not show significant changes, sympathetic nerves were significantly reduced in chronic pancreatitis and pancreatic cancer [17]. Interestingly, in PDAC patients with PNI and/or endoneural invasion, the nerve cholinergic content was reduced or nearly diminished [17].

Concerning the alterations in neuromorphology and function, we often use the term “neuroplasticity” [18]. There is no difference in the neural remodeling between different PDAC stages, emphasizing the relative global characteristics of this phenomenon [17,19]. It is primarily manifested as nerve hypertrophy and density changes in and adjacent to the pancreatic parenchyma in PDAC. The former exists in both interlobular and intralobular nerves [20,21,22]. However, contrary to the traditionally held belief of increased nerve density [23], studies have indicated that the nerve density in PDAC, especially in the center of the tumor, is significantly lower than normal pancreas [24,25]. Possible reasons include but are not limited to the following points: (i) aggressive tumor cells expand from the center to the periphery progressively and then crush the nerves [13]; (ii) the immunohistochemical methods used in most studies cannot show the loss of slender nerve endings [24]; and (iii) deviations caused by differences in marker selection, detection methods, and measurement accuracy cannot be ruled out.

## 3. Dorsal Root Ganglion: Responsible for PNI

The dorsal root ganglion is an inflated nodule of the dorsal root of the spinal cord near the inner side of intervertebral foramen. It is responsible for receiving nerve impulses from receptors and transmitting them to the spinal cord, becoming the starting point of the pain conduction pathway. In addition to acting as a conduction point, DRG is also an essential part of the crosstalk between cancer cells and neurons. Researchers brought about a series of animal models to simulate the TME of PDAC (Table 1).

### 3.1. The Effects of DRG on PDAC Invasiveness during PNI

The dorsal root ganglion and its corresponding spinal cord are extremely sensitive to pancreatic disease. In the PKC mouse model, an autochthonous model of PDAC, cells derived from the pancreas invaded the spinal cord along the sensory neurons and migrated to the lower thoracic and upper lumbar regions as early as the PanIN2 stage [5,31]. In the tumor microenvironment (TME), the DRG undergoes structural and functional reorganization. Structurally, it shows an increase of synapses. Functionally, it actively participates in the occurrence and development of PNI through the expression and secretion of a variety of cytokines, including neurotrophins (NT), glial cell line-derived neurotrophic factors (GDNF), axon guiding molecules, chemokines, and adhesion molecules (Table 2 and Table 3). In the following part, we will review the GDNF family in detail.

Recently, an increasing number of studies have demonstrated the importance of GDNF. GDNF family members bind to GDNF family receptor-α (GFRα) coreceptors and subsequently recruit RET (rearranged during transfection) for dimerization. The molecular mechanism induces polarization and actin redistribution, and enhances the directional movement of PDAC cells by stimulating the downstream signal pathways, including ERK, JNK, p38 MAPK, and PI3K/AKT [40].

The increased RET receptors are found in aggressive cancers, such as PDAC, non-small-cell lung cancer, and papillary thyroid cancer [62]. Besides the increased expression level of RET, different RET isoforms and polymorphic variants are important in cancer development and migration. RET9 and RET51 are the two widely accepted isoforms, of which the latter plays a predominant role in the PDAC invasiveness [40]. Moreover, Capan-2, and MIA PaCa-2 with RET polymorphism are more sensitive to GDNF stimulation than those with wild-type RET, meaning a more profound influence on the proliferation and invasion of PDAC cell lines [33].

### 3.2. The Metabolic Effects of DRG on PDAC Proliferation

Recent studies have shown the potential metabolic impact of DRG on PDAC [63,64]. Due to the lack of blood supply and oxygen, PDAC develops multiple strategies to meet its nutritional needs. Serine is responsible for protein synthesis, and an impaired serine biosynthesis pathway in approximately 40% of PDAC cells means their dependence on exogenous serine. When faced with serine shortage, PDAC cells would undergo selective serine-codon ribosome pausing and enhance NGF translation, thereby increasing tumor innervation and promoting the serine release from DRG [65].

## 4. Schwann Cell: A Positive Role in PNI

Schwann cells are an essential component of peripheral nerves and are primarily divided into three subtypes. Myelin Schwann cells wrap around the axons and myelinate them, forming a node of Ranvier that is necessary for the saltatory transmission of action potentials. The Remak Schwann cells are attached to unmyelinated axons and distributed along them, thus constituting Remak bundles [66]. Studies have demonstrated that myelin Schwann cells mainly myelinate g-motor axons, Aα, Aβ, and Ad-sensory axons, while Remak cells are associated with C fibers and autonomic nerves.

### 4.1. Activated Schwann Cells in the TME of PDAC

Interestingly, Schwann cells are widely distributed in pancreatic cancer tissues. Similar to neurogenic inflammation [16], Schwann cells exist around the lesion as early as the pancreatic intraepithelial neoplasia occurs, which is considered a typical manifestation before the development of invasive cancer [67]. Instead of a continuous connection with nerves, Schwann cells, labeled with specific markers S100 and glial fibrillary acidic protein (GFAP), represent a “nerve-independent, stromally distributed” pattern in human pancreatic cancer and two commonly used PDAC transgenic mouse models, the KC mice (p48-Cre/LSL-Kras^G12D^) and KP mice (p48-Cre;LSL-Kras^G12D^;Trp53^−/−^) [7]. Schwann cells can also be observed in direct contact with the epithelial compartment [47]. However, the origin of Schwann cells in PDAC is still controversial. Some researchers believe that it might be due to the ongoing tissue remodeling in pancreatic cancer [7].

In the TME, Schwann cells can be activated by hypoxia, tumor cells, and factors secreted by T cells, such as IL-6 [67], and then undergo a series of phenotypic and functional reprogramming [55]. This process is similar to reversible dedifferentiation in the case of nerve injury and is accompanied by the up-regulation of a variety of neurotrophins.

### 4.2. The Multiple Roles of Intrapancreatic Schwann Cells in PNI

The traditional view is that cancer cells play a dominant role in PNI, while nerve fibers and their surrounding glia cells are just “victims” of the malignant process (Figure 1). However, the latest research has broken through the outdated concept. Astonishingly, research has illustrated that before PDAC cells began to migrate to the neurons, Schwann cells had already started migrating to the pancreatic cancer cells [7]. The three-dimensional tumor cell sphere also allows us to observe the phenomenon more intuitively: GFAP^+^ Schwann cells destroy the sphere and enhance tumor invasion by promoting the separation of individual cells, forming linear cell chains, and inserting cells to destroy cancer cell contacts [55].

Clinical data indicate that high level of intratumoral Schwann cell marker GFAP/S100 could be applied as an independent prognostic factor for the poor survival of PDAC patients [47]. The specific molecular mechanisms are as follows. Schwann cells can induce PDAC epithelial-to-mesenchymal transition (EMT) to help them gain migratory properties [68]. Moreover, Schwann cells secrete chemical attractants and degrade the matrix, thereby providing a transmission route for the early-stage migration of tumors. The pathways involved include: NGF-TrKA-p75NTR, L1CAM-MAPK, L1CAM-STAT3-MMP2/9, CXCL12-CXCR4/7, and laminin–integrin [7,11,57] (Table 2 and Table 3).

## 5. The Pain Initiation and Conduction in PDAC Patients

Abdominal pain, which is associated with poor prognosis, occurs in approximately 80% of PDAC patients [69]. The source of the pain is complicated and remains unclear. Recently, increasing evidence has been supporting the role that neurogenic inflammation plays in pain initiation. Several studies have confirmed the presence of intrapancreatic neurogenic inflammation as early as PanIN2 stage [5]. In the TME, the axons of the DGR induces the release of substance P and CGRP by up-regulating transient receptor potential cation channel subfamily V member 1 (TRPV1) [70]. These factors mediate neurogenic inflammation, leading to the stimulation of nociceptive nerve fibers [71]. The action potential then goes along the spinal cord and activates the secondary neurons, thereby transmitting pain signals to the brain (Figure 2).

Glial cells undoubtedly play a unique role in neuropathic pain transmission. However, it is worth noting that the role of glial cells varies greatly according to different conditions. For example, in mouse sciatic nerve injury models and alcohol-induced neuropathic pain models, Schwann cells have been shown to promote pain through NOX1-dependent hydrogen peroxide release [72,73]. On the contrary, the secretion of IL-6 by Schwann cells in the microenvironment of PDAC is thought to be related to the decreased abdominal mechanical sensitivity [67]. Although the specific mechanism has yet to be further explored, these studies remind us that we should look at the relationship between glial cells and pain from a global perspective.

In this process, macrophages are recruited into the perineural space under the action of chemokines. The oxidative stress storm they produce plays an essential role in initiating and maintaining pain [72]. On the one hand, macrophages can directly activate TRPV1 on nociceptors by releasing reactive oxygen species and other substances [74]; on the other hand, macrophages indirectly exert their influence on neuropathic pain by activating glial cells in nerve roots [75,76,77]. Similar findings have been preliminarily confirmed in pancreatic cancer [78].

## 6. Autonomic Nerves: Interacting with Immunity

Autonomic nerves are widely distributed in pancreatic cancer. They play an irreplaceable role in the development of PDAC. In recent years, researchers have demonstrated that the interaction between autonomic nerves and tumor cells had two sides.

### 6.1. Sympathetic Nerve: Adverse Effects Depending on Different Conditions

Epidemiological studies have shown that there is extensive crosstalk between neuropsychological stress and tumor development [79]. As an essential way for stress-mediated tumor progression, sympathetic innervation in pancreatic cancer has been confirmed in human and animal models. In most cases, stress/sympathetic innervation is believed to promote the occurrence and development of PDAC. Researchers have revealed that an increase in β-adrenaline caused by stress can activate the NGF-BDNF/Trk pathway in a β2-adrenergic receptor (ADRB2)-dependent mode, thereby promoting acinar-to-ductal metaplasia (ADM), which is considered to be the first step of PDAC tumorigenesis [80]. Norepinephrine can also promote PNI by increasing the phosphorylation level of STAT3 [81]. In addition, Eng et al. noted that coldness-induced stress could enhance norepinephrine and activate β-adrenergic receptors, leading to the resistance to cisplatin and paclitaxel [82].

However, a study on the enriched environment (EE) showed that moderately positive stimulation could increase the infiltration and vitality of natural killer cells through the mediation of sympathetic nerves, thus presenting a phenotype of tumor inhibition [29]. EE is an experimental paradigm characterized by more living space and more complex social interactions and is believed to provide mice with sensory, cognitive, motor, and social stimulation, without making them feel painful.

### 6.2. Parasympathetic Nerve: A Double-Edged Sword in PDAC

Parasympathetic nerves exist in various solid tumors and play diverse roles in different tumors [83,84,85,86]. Contrary to prostate cancer, parasympathetic nerves inhibited the occurrence of Kras mutant pancreatic cancer [87]. Renz BW et al. believed that the opposite effect might contribute to the location specificity of the nerve input [27]. In PDAC, enhanced cholinergic signals could directly inhibit MAPK/EGFR, and PI3K/AKT pathways which were mediated by type 1 muscarinic receptors (CHRM1), but also could indirectly suppress tumor progression by inhibiting tumor stem cell transformation [27]. This is consistent with the possible antitumor effect of the vagus nerve reported in previous clinical studies.

However, a study has shown that acetylcholine could reduce the recruitment of CD8^+^ T cells, inhibit the differentiation of T cells into Th1 cells, and ultimately assist tumors in achieving immune escape [88]. In addition, clinical studies have shown that parasympathetic neurogenesis is closely related to tumor sprouting and poor prognosis in PDAC [89]. The expression of muscarinic acetylcholine receptor 3 is significantly correlated with high grade, more lymph node metastasis, and shorter overall survival of PDAC patients [90].

From these seemingly contradictory results, it is not difficult to see that the role of autonomic nerves in PDAC is complicated, especially when immune factors are involved. The exact functions of autonomic nerves and their corresponding neurotransmitters need to be further studied in the context of the TME components and disease stages.

## 7. Macrophages: All-Rounder in Tumor Progression

Stroma in pancreatic cancer is abundant, reaching up to more than 80% of the total tumor volume. In addition to various extracellular matrix components, there are also multiple kinds of cells in the stromal compartment, including cancer-associated fibroblasts (CAFs), pancreatic stellate cells (PSCs), tumor-associated macrophages (TAMs), myeloid-derived suppressor cells (MDSCs), and regulatory T cells [91]. They possess unique characteristics in the TME [92], and secrete cytokines as “signal transporters” to participate in the tumor–neural interaction. We will focus on TAMs in the following part.

TAMs are derived from circulating monocytes and resident tissue macrophages. Cytokines produced by tumors actively recruit them. Studies have indicated that Hif-1α could improve the expression and secretion of granulocyte–macrophage colony-stimulating factor (GM-CSF) by combining with the GM-CSF promoter, thereby increasing the infiltration of macrophages [49].

In pancreatic cancer, TAMs can increase the invasiveness, EMT, and metastasis of tumor cells [93,94,95], meanwhile acting as a “guide” and “mediator” for migration, thus promoting PNI. Compared with normal nerves, there are more macrophages around invaded nerves [96]. Real-time fluorescence microscopy imaging has shown that macrophages were recruited to the forefront several hours before the occurrence of PNI. The recruited macrophages assisted the polarization and migration of tumor cells through the GDNF/RET pathway without changing the direction of tumor migration [34].

In addition, macrophages can act on the stromal compartment of PDAC to promote tumor progression indirectly. A notable finding has shown that macrophages could stimulate CAFs to produce leukemia inhibitory factors (LIF) and SLIT2, thereby inducing the plasticity of the DRG, and promoting the nerve invasion of cancer cells through JAK/STAT3/AKT and N-cadherin/β-catenin pathways [41,48]. Moreover, activated PSCs by TAMs can increase MMP-2, MMP-9, and NGF expression, thereby increasing the proliferation and migration of PDAC cells and the occurrence rate of PNI [97].

## 8. CNS Alterations in PDAC and Chronic Pancreatitis

Cognitive abnormity is common in patients with pancreatic cancer. Previous studies have attributed the abnormality to treatment-related toxicity and psychological stress after diagnosis, with little attention to cancer itself [98]. Recent studies have shown that the inflammatory mediators produced by tumor cells could promote the infiltration of circulating immune cells, mainly the neutrophils, to the meninges around the hippocampus via the CCR2–CCL2 axis, leading to a decline in cognitive levels and anorexia [6]. In addition, considering that the extracellular vesicle (EV) can cross the blood–brain barrier [99], it may also become one of the potential mechanisms that carry PDAC-derived miRNAs to CNS.

Cancer-associated pain is also an important issue, affecting the life quality of PDAC patients. The widely recommended analgesia approaches include systematic analgesic medications and celiac plexus/ganglia neurolysis (Table 4). The latter is usually applied when analgesic therapy is ineffective or brings about unacceptable adverse effects. To some extent, the outcomes of the following clinical trials are inconsistent. This may be due to insufficient sample size, differences in physician skills, and changes in endoscopic equipment and painkillers in different times. Prospective large-scale, multicenter clinical trials are needed to verify the effectiveness of neurolysis.

However, even with developed analgesics and celiac plexus/ganglia neurolysis visualization, pain in some patients cannot be fully relieved [108]. CNS alteration and accompanying central sensitization appear to contribute to the intractable pain and relevant studies may inspire the development of pain management.

Currently, there are few studies on CNS alterations in pancreatic cancer patients. Considering that there are many similarities between chronic pancreatitis (CP) and PDAC, we mainly enumerate CNS changes in CP for reference, including central sensitization, damage to the descending inhibitory pathway, and changes in resting brain activity [18].

Central sensitization is essentially a phenomenon of synaptic plasticity, which is believed to be caused by neuroinflammation in the peripheral and central nervous system [109]. At present, the regions of CNS involved in pain-related remodeling mainly contain the anterior cingulate cortex (ACC), thalamus, amygdala, insular cortex, prefrontal cortex, and spinal dorsal horn (Figure 2). Among them, ACC is reported to accept pain signals from the thalamus and amygdala [110,111]. Studies have shown that in CP patients, the transmission of the glutamatergic excitatory synapse located in the bilateral insula and ACC was enhanced, accompanied by a significantly shorter latency of the early evoked potential (EP), which was considered to be related to pain-related anxiety and alarm [112,113,114].

Recent studies have shown that the descending inhibitory pathway influenced the maintenance of CP-related pain greatly [115,116], and periaqueductal gray (PAG) is reported to be a necessary component of it. It transmits excitability to the rostral ventral medulla (RVM) and then to the spinal dorsal horn, thus forming the PAG–RVM-spinal descending inhibition pathway [117,118]. In the DBTC-induced rat model of CP, the reduction of glutamatergic synaptic strength mediated by the pre- and post-synaptic mechanisms in the ventrolateral periaqueductal gray (vlPAG) could be found, leading to abdominal hypersensitivity [119].

## 9. Clinical Transformation: Cancer Neural Therapy

Due to the negative effects of PNI, people are trying to develop some methods to target cancer-nerve crosstalk. The strategies include β-blockers, surgical resection of peripheral nerves, and electrical stimulation [120,121]. Propranolol, one of the most typical β-blockers, was shown to inhibit tumor invasiveness in mice PDAC models [122]. However, there is no strong clinical evidence to support its effectiveness in human PDAC. Moreover, surgical approaches cannot selectively remove a specific component of the nerve, becoming one of the biggest obstacles to cancer neural treatment.

In recent years, adeno-associated virus (AAV) has developed rapidly and has become a vehicle for cancer neural treatment. Research has confirmed the safety and efficacy of AAV retrograde intrapancreatic administration, which was more effective than systemic delivery [123]. A single-stranded DNA-packaged polyplex micelle was shown to work as AAV-inspired compact vector, so as to target stroma-rich pancreatic cancer [124]. In addition, modified AAV vectors can carry transgenes downstream of nerve fiber type-specific promoters to target nerve in a selective manner [120]. Despite the lack of relevant clinical trials, AAV is undoubtedly a promising vehicle for cancer neural therapy and needs to be further explored.

## 10. Discussion

As an important disseminating route of PDAC, the nervous system has attracted worldwide attention. In addition to providing a physical metastasis pathway, the sensory and autonomic nerves secrete different neurotransmitters to interact with cancer cells. Stromal cells in TME also undergo functional transformation and regulate multiple signal pathways, thereby promoting the development of PNI.

Clinically, PNI is an independent predictor of poor prognosis and abdominal cavity reoccurrence. The presence of PNI after preoperative gemcitabine-based chemoradiation therapy indicates the tumor aggressiveness and its resistance to gemcitabine [125]. Once PNI is discovered, postoperative chemoradiation and a drug combination strategy are preferred. In addition, special attention should be paid to the results of abdominal cavity imaging during the reexamination.

However, the research of PNI is still limited by the following facts. Because of the hidden symptoms of PDAC, diagnosed patients are usually at a relatively late stage, making it almost impossible to observe the nerve innervation of PanIN; nerves crushed by aggressive cancer cells worsen the situation. On the other hand, although genetically engineered mouse models can exhibit some characteristics of PanIN and early-stage PDAC, the extent to which animal models can stimulate human pancreatic cancer remains a problem, especially considering the anatomic differences of intrapancreatic innervation between mouse and human. Better animal models and more extensive clinical trials are needed to help us deepen the understanding of PNI, better explain its relationship with low survival rate of PDAC, and potentially bring hope to patients with severe PDAC-associated pain.

## 11. Conclusions

PNI is an outstanding clinical feature of PDAC, which is associated with poor prognosis and reoccurrence. Doctors can better design the treatment plan of patients according to whether there is PNI. The tumor–nerve interaction in PDAC is quite complex. It is a problem of more than the cancer and nerve cells themselves but involves diverse components within the cancer stromal compartment, such as CAFs, TAMs, and PSCs. These factors interact with each other and form a vicious circle, leading to the malignant behavior of PDAC. In addition, the complex interplay may also help trigger and maintain pancreatic cancer-associated pain. We look forward to the development of this field to help us unravel the specific mechanisms behind PNI so as to develop new targets and effective strategies for treating pancreatic cancer.

## Figures and Tables

**Figure 1 cancers-13-04594-f001:**
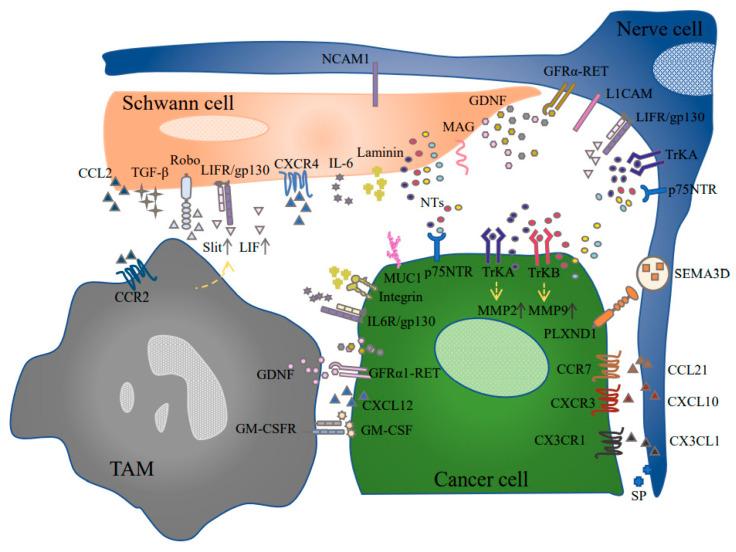
The complicated crosstalk between different types of cells in TME. Various signaling molecules participate in the interaction, thereby transforming the cell structure and function and facilitating the malignant development of PDAC. GDNF, glial cell line-derived neurotrophic factor; GFRα-RET, glial cell line-derived neurotrophic factor family receptor-α-rearranged during transfection; GM-CSF, granulocyte–macrophage colony-stimulating factor; GM-CSFR, granulocyte–macrophage colony-stimulating factor receptor; IL6R/gp130, interleukin-6 receptor/glycoprotein 130; L1CAM, L1 cell adhesion molecule; LIF, leukemia inhibitory factor; LIFR/gp130, leukemia inhibitory factor receptor/glycoprotein 130; MAG, myelin associated glycoprotein; MUC1, mucin 1; NCAM1, neural cell adhesion molecule; NT, neurotrophin; p75NTR, p75 neurotrophin receptor; PLXND1, plexin-D1; Robo, Roundabout; SEMA3D, semaphorin 3D; SP, substance P; TAM, tumor-associated macrophage; TrKA, tropomyosin receptor kinase A; TrKB, tropomyosin receptor kinase.

**Figure 2 cancers-13-04594-f002:**
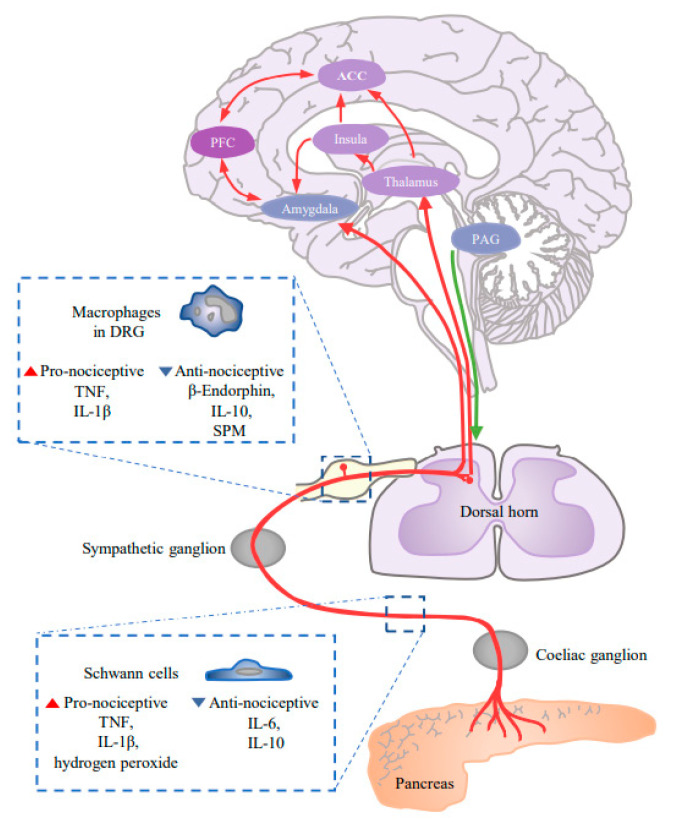
The alterations of pain conduction in PDAC patients. In the peripheral nervous system, local neurogenic inflammation in PDAC initiates the pain conduction. Schwann cells and macrophages regulate the process by the release of multiple pro-nociceptive factors. In the CNS, several pain-related brain regions undergo a series of reorganization and remodeling of synapses, contributing to the pain sensitization in PDAC patients. ACC, anterior cingulate cortex; PAG, periaqueductal gray; PFC, prefrontal cortex; SPM, specialized pro-resolving mediator.

**Table 1 cancers-13-04594-t001:** Reported animal models in studying the crosstalk between PDAC and nerves.

Class of PDAC Models	Representatives	Advantages	Disadvantages
Transplanted tumor models [26]	Subcutaneous transplantation Sciatic nerve injection	Common Low cost Short reproduction cycle	Cannot simulate the anatomical relationship between nerves and tumors
Genetically engineered mouse models [12,13]	p48-Cre/LSL-Kras^G12D^ p48-Cre;LSL-Kras^G12D^;Trp53^−/−^ PlxnD1^f/f^ ADV-CRE^+/−^ LSL-KRAS^G12D/+^;LSL-TP53^R172H/+^;PDX-1-CRE^+/+^;ANXA2^−/−^ GFRα1^+/−^ heterozygote mice	Better reflect the dynamic changes of pancreatic lesions, especially at the early stage	High cost Long reproduction cycle
Surgical intervention [27]	Selective vagotomy of the hepatic vagal branch	Study the effects of specific vagal branch	Infection risk Difficulty in operation
Chemical removal [5,28,29]	6-hydroxydopamine injection TRPV agonists: capsaicin, and resiniferatoxin	Ablate C fibers and TRPV^+^ afferent sensory nerve fibers selectively	Unclear collateral effects
Others [30]	Herpes simplex virus	Neurophilicity	Toxicity

**Table 2 cancers-13-04594-t002:** Typical nerve-related factors involved in PNI.

Class of Molecules	Components	Reported Mechanisms and Their Roles in PNI	References
NGF family	NGF BDNF NT3 NT4/5	Mainly secreted by the tumor and surrounding stromal cells. Each neurotrophic factor binds to a specific high-affinity receptor TrK and a low-affinity receptor p75NTR, thereby promoting tumor cell migration to the nerve.	[4,7,8,9,10,32]
GDNF family	GDNF Neuturin Artemin Persephin	GDNF, neuturin, artemin and persephin combine to GFRα1/α2/α3/α4, respectively, and then interact with RET to activate downstream signal pathways, including ERK, JNK, p38 MAPK and PI3K/Akt, thus enhancing migration capacity.	[4,12,33,34,35,36,37,38,39,40]
Axon Guidance Molecules	SLIT ROBO	The SLIT secreted by CAFs regulates Schwann cells N-cadherin/β-catenin through the SLIT-ROBO pathway, thereby promoting the migration.	[41,42,43]
	SEMA3D PLXND1	SEMA3D secreted by neuro-exocytosis acts on the PLXND1 of tumor cells, thereby enhancing invasiveness, accompanied by an increase in nerve density.	[13,42,44]
Others	SP	Increased SP secretion acts on the NK1-R^+^ cell subgroup in PDAC and promotes STAT3/JAK2 and MAPK/ERK pathways.	[26,28,45]

BDNF, brain-derived neurotrophic factor; CAF, cancer-associated fibroblast; GDNF, glial cell line-derived neurotrophic factor; GFR, glial cell line-derived neurotrophic factor family receptor; NGF, nerve growth factor; NK1-R, neurokinin 1-receptor; NT, neurotrophin; p75NTR, p75 neurotrophin receptor; PLXND1, plexin-D1; RET, rearranged during transfection; Robo, Roundabout; SEMA3D, semaphorin 3D; SP, substance P; TrK, tropomyosin receptor kinase.

**Table 3 cancers-13-04594-t003:** Cytokines, chemokines, and cell-surface adhesion molecules involved in PNI.

Class of Molecules	Components	Reported Mechanisms and Their Roles in PNI	References
Cytokines	TGF-β TGF-βR	TGF-β can activate the Smad signaling pathway of tumor cells, enhancing invasion and metastasis.	[46]
	IL-6 IL6R & gp130	IL-6 can activate the tumor STAT3/AKT pathway.	[47,48]
	GM-CSF GM-CSFR	Under hypoxia induction, tumor cells secrete more GM-CSF, attracting Schwann cells to migrate to them.	[49]
Chemokines	CCL21/CXCL10 CCR7/CXCR3	CCL21/CXCL10 can bind to CCR7/CXCR3 respectively, promoting the migration capability through the AKT pathway. And they play a role in nociceptive hypersensitivity and neural remodeling without affecting tumor immune filtration.	[50]
	CXCL12 CXCR4	CXCL12 acts on CXCR4, enhancing the chemical attraction to Schwann cells.	[11]
	CX3CL1 CX3CR1	CX3CL1 acts on CX3CR1, promoting the adhesion of PDA cells to peripheral nerves.	[51,52]
	CCL2 CCR2	The activated Schwann cells secrete CCL2 and recruit more macrophages, which play an important role in tumor invasion.	[53,54]
Cell-surface adhesion molecules	NCAM1	Mainly exists on the Schwann cells and promotes tumor migration.	[55]
	L1CAM	As a molecule that regulates the adhesion and migration of neurons, it’s associated with higher PNI rate and poor prognosis, potentially because of its ability to activate MAPK pathway and promote the expression of MMP2/9.	[56]
	Laminin Integrin	The laminin can attract tumor cells by activating the MAPK/ERK pathway and increase their MMP2/9 expression. Extracellular matrix laminin can also be combined with integrin, promoting tumor invasion.	[57,58,59,60]
	MUC1 MAG	Tumor cells and Schwann cells interact with each other through MUC1-MAG combination to promote PNI of PDAC.	[61]

GM-CSF, granulocyte-macrophage colony-stimulating factor; GM-CSFR, granulocyte-macrophage colony-stimulating factor receptor; IL6R/gp130, interleukin-6 receptor/ glycoprotein 130; L1CAM, L1 cell adhesion molecule; MAG, myelin associated glycoprotein; MUC1, mucin 1; NCAM1, neural cell adhesion molecule.

**Table 4 cancers-13-04594-t004:** Clinical trials of pain management in human pancreatic cancer.

Team	Design	Assessment of Pain	Primary Outcome
Gilbert et al. [100]	Double-blind, randomized trial	Pain intensity QOL Opioid consumption	NCPB improved pain relief compared with optimized systemic analgesic therapy alone, without affecting survival.
Michael et al. [101]	Retrospective trial	Not mentioned	EUS-guided direct celiac ganglion block/neurolysis was safe. Alcohol injection into ganglia appeared to be effective.
Gil et al. [102]	Retrospective analysis of prospective database	VAS score for pain Narcotic use	Visualization of celiac ganglia with direct injection was the best predictor of response to EUS-CPN.
S. Doi et al. [103]	Randomized multicenter trial	Abdominal pain intensity score	EUS-CGN was significantly superior to conventional EUS-CPN in pain relief.
Larissa et al. [104]	Retrospective, case-control trial	Opioid consumption	EUS-CN was associated with longer survival compared with non-EUS approaches, and those who received EUS-CPN had longer survival than EUS-CGN.
Ji Young Bang et al. [105]	Randomized controlled trial	the EORTC Quality of Life Questionnaire pancreatic cancer module and core questionnaire the Brief Pain Inventory-Short Form VAS score for pain	EUS-RFA provided more pain relief and improved QOL than EUS-CPN.
Michael et al. [106]	Randomized, double-blind trial	Pain intensity and character Morphine response Opioid use converted to daily oral morphine equivalents	Compared with EUS-CPN, EUS-CGN was associated with reduced median survival time without improving pain, particularly for patients with non-metastatic disease.
Yoshihide Kanno et al. [107]	Prospective randomized control trial	VAS score for pain VAS score for QOL Opioid use converted to daily oral morphine equivalents	EUS-CPN did not appear to improve pain, QOL, or opioid consumption compared with those who did not undergo EUS-CPN and medicated with oxycodone/fentanyl.

CN, celiac neurolysis; EORTC, European Organization for Research and Treatment of Cancer; EUS-CGN, EUS-guided celiac ganglia neurolysis; EUS-CPN, EUS-guided celiac plexus neurolysis; EUS-RFA, EUS-guided radiofrequency ablation; NCPB, neurolytic celiac plexus block; QOL, quality of life; VAS, visual analog scale.

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
