# Peer review of "Perineural Invasion and Associated Pain Transmission in Pancreatic Cancer"

_cancers, 2021, doi:10.3390/cancers13184594_

Round 1
Reviewer 1 Report
In this review Wang et al. give a good overview over the very complex field of perineural invasion in pancreatic cancer.
I have some minor issues:
- Language should be checked.
- What are the clinical conclusions of this paper? How can the knowledge of PNI be used for the treatment of PDAC patients?
- There is still a lot of research to be performed on this field. What are the next steps to close the gap between theoretical knowledge and clinical use?
- Is there any difference in PNI between different pancreatic tumors, like PDAC or IPMN etc.?
Author Response
Dear reviewer,
Thank you very much for your time involved in reviewing the manuscript and your very encouraging comments on the merits
We also appreciate your clear and detailed feedback and hope that the explanation has fully addressed all of your concerns. In the letter, we discuss each of your comments individually along with our corresponding responses.
To facilitate this discussion, we first retype your comments in italic font and then present our response to the comments.
Comment 1:
Language should be checked.
Response 1:
Thank you for the detailed review. We have carefully and thoroughly proofread the manuscript to correct all the grammar and typos.
Comment 2:
What are the clinical conclusions of this paper? How can the knowledge of PNI be used for the treatment of PDAC patients?
Respond 2:
Thank you for pointing these problems out. As a unique clinical feature of PDAC, 80-100% of PDAC patients will develop PNI, which is significantly associated with poor prognosis and low quality of life. Moreover, PNI is an independent predictor of reoccurrence in the abdominal cavity. We have revised the corresponding conclusion part on Page 12, lines 401-403.
The presence of perineural invasion after preoperative gemcitabine-based chemoradiation therapy indicates the tumor aggressiveness and its resistance to gemcitabine. Once PNI is discovered, postoperative chemoradiation and drug combination strategy are preferred. In addition, special attention should be paid to the results of abdominal cavity imaging during the reexamination. Please see the revised part on Page 11-12, lines 378-383.
Comment 3:
There is still a lot of research to be performed on this field. What are the next steps to close the gap between theoretical knowledge and clinical use?
Response 3:
We appreciate your valuable suggestions. As you said, there is still a certain gap between theoretical research and clinical use. The reliable clinical trials of PNI-target therapy are rare. From our point of view, to close the gap, vectors for targeted drug delivery, such as AAV mentioned on Page 11, lines 355-371, should be developed. In addition, prospective large-scale, multicenter clinical trials are needed to evaluate the safety and efficacy of treatment strategies.
Comment 4:
Is there any difference in PNI between different pancreatic tumors, like PDAC or IPMN etc.?
Response 4:
Thanks a lot for your professional comments. At present, the research on the perineural invasion mechanism of IPMN is very limited. According to some existing clinical studies, the perineural invasion rate of IPMN is lower than that of PDAC, which might be an important factor for the better prognosis of invasive IPMN compared with standard PDAC.
We would like to take this opportunity to thank you for all your time involved and this great opportunity for us to improve the manuscript. We hope you will find this revised version satisfactory.
Sincerely
Xiaoping Zou

Reviewer 2 Report
In this manuscript entitled “Perineural invasion and associated pain transmission in pancreatic cancer”, Wang et al present a comprehensive overview of the current knowledge on PDAC-related pain. This research field, while quite active, still has to fully understand the origin of pain in PDAC, as well as the roles played by nerves, inflammation, CNS, chemotherapies… in the associated processes. As such, this manuscript gives a good account of what is known, and of the next challenges still to be addressed, while providing relevant tables and schemes.
Comments:
Reference 37 (line 174) appears to be incorrect: ref 36 should be mentioned instead (ref 37 is only a comment on ref 36, with no listed authors).
In table 4, the authors give several references about clinical trials. There is however no indication as to whether these trials are still ongoing or not. Moreover, these references are not discussed in the manuscript. What are the ongoing clinical trials trying to target nerves, inflammation, or molecular mechanisms described by the authors? If such studies exist, some examples should be provided and discussed by the authors.
Author Response
Dear reviewer,
Thank you very much for your time involved in reviewing the manuscript and your very encouraging comments on the merits
We also appreciate your clear and detailed feedback and hope that the explanation has fully addressed all of your concerns. In the letter, we discuss each of your comments individually along with our corresponding responses.
To facilitate this discussion, we first retype your comments in italic font and then present our response to the comments.
Comment 1:
Reference 37 (line 174) appears to be incorrect: ref 36 should be mentioned instead (ref 37 is only a comment on ref 36, with no listed authors).
Response 1:
Thank you for the detailed suggestion. We have corrected this error in the manuscript. Please see Page 6, line 174.
Comment 2:
In table 4, the authors give several references about clinical trials. There is however no indication as to whether these trials are still ongoing or not. Moreover, these references are not discussed in the manuscript. What are the ongoing clinical trials trying to target nerves, inflammation, or molecular mechanisms described by the authors? If such studies exist, some examples should be provided and discussed by the authors.
Response 2:
Thank you for pointing these problems out. The first study from Gilbert's team in Table 4 was not found on major clinical trial websites due to its long history. And the remaining four prospective clinical studies have all been completed, while the registration and completion of the other three retrospective trials were not mentioned.
To some extent, the outcomes of these clinical trials in Table 4 are inconsistent, and the possible reasons include the following three parts: (1) insufficient sample size; (2) differences in surgeon skills; (3) differences in endoscopic instruments and analgesic drugs in different times. The clinical trials mentioned in Table 4 may help us understand the possible relationship between PNI, prognosis and pain in a dialectic manner. It is a field that needs to be further studied in large-scale, multicenter clinical trials. Please see Page 9, lines 315-319.
At present, the cancer neural therapy is still in the stage of animal experiments, and reliable clinical trials are rare. Development of targeted drug delivery and objective evaluation of the safety and efficacy of related treatment strategies are needed to promote the clinical transformation of cancer neural therapy in PDAC. We have added the current condition of clinical transformation in PDAC neural therapy. Please see Page 11, lines 355-371.
We would like to take this opportunity to thank you for all your time involved and this great opportunity for us to improve the manuscript. We hope you will find this revised version satisfactory.
Sincerely
Xiaoping Zou

Reviewer 3 Report
This is a well-written and extensive review of PNI in PDAC.
I have only one minor remark:
There is no mention of any clinical trials of nerve-directed therapy in PDAC, despite numerous preclinical data. Is that correct or not ? In any case this should be mentioned in the discussion or as a future aim.
Author Response
Point-by-Point Response
Dear reviewer,
Thank you very much for your time involved in reviewing the manuscript and your very encouraging comments on the merits
We also appreciate your clear and detailed feedback and hope that the explanation has fully addressed all of your concerns. In the letter, we discuss each of your comments individually along with our corresponding responses.
To facilitate this discussion, we first retype your comments in italic font and then present our response to the comments.
Comment 1:
There is no mention of any clinical trials of nerve-directed therapy in PDAC, despite numerous preclinical data. Is that correct or not? In any case this should be mentioned in the discussion or as a future aim.
Response 1:
Thank you for pointing this out. At present, the cancer neural therapy is still in the stage of animal experiments, and reliable clinical trials are rare. Development of targeted drug delivery and objective evaluation of the safety and efficacy of related treatment strategies are needed to promote the clinical transformation of cancer neural therapy in PDAC. We have added the current condition of clinical transformation in PDAC neural therapy. Please see Page 11, lines 355-371.
We would like to take this opportunity to thank you for all your time involved and this great opportunity for us to improve the manuscript. We hope you will find this revised version satisfactory.
Sincerely,
Xiaoping Zou